## Comment

 

**Author for correspondence:**
Keith M. Kendrick
e-mail: k.kendrick.uestc@gmail.com

# Response to 'Sheep recognize familiar and unfamiliar human faces from two-dimensional images'

## Keith M. Kendrick

The Clinical Hospital of Chengdu Brain Science Institute, MOE Key Laboratory for NeuroInformation of Ministry of Education, Center for Information in Medicine, University of Electronic Science and Technology of China, Chengdu, People's Republic of China

KMK, 0000-0002-0371-5904

The comment by Towler *et al.* [1] questioning Knolle, Goncalves and Morton's [2] claims that their findings represent evidence for sheep having advanced human-like face-recognition abilities makes some good points, and the limited number of animals involved and face stimuli together with the lack of direct comparisons between sheep and humans clearly make conclusions based only on their experiments somewhat preliminary. However, the main contribution of the original paper is to provide further evidence that sheep possess an expert face-recognition ability that has some similarities with that of humans [2]. As such, it builds upon a large body of previous behavioural and neural evidence for face-recognition expertise in sheep, including involvement of the temporal cortex and right hemisphere dominance, produced over a period of over 30 years and which is not fully discussed [3]. There are some key points that both the original paper and the comment by Towler *et al.* might have considered.

Both the original paper and Towler *et al.*'s comment fail to raise the obvious issue that just as humans are better at recognizing human compared to sheep faces (see [4], for example) so are sheep better at recognizing sheep compared to human faces. In terms of sheep facial recognition several previous publications have reported weaker evidence for configural encoding of human compared with sheep faces [5,6]. Indeed, at the neural level, different populations of cells in the temporal cortex respond to sheep as opposed to human faces [7], although some overlap can occur in the case of highly familiar humans (see [8]). Similarly, in human prosopagnosia patients there is evidence that neural processing of human faces may be distinct to some extent from that of the faces of other species [4,9]. Thus, if a true comparison is going to be made between the relative expertise of humans and sheep, then both human and sheep face stimuli should be used. It is also

important to note that sheep visual acuity in the frontal field is somewhat inferior to that of humans, and this too may contribute to performance differences.

The question of whether face stimuli used in experiments are processed simply as pictures or as a representation of a specific individual is clearly of great importance, since many different species have been shown to be capable of making very sophisticated visual discriminations. This is why the Knolle *et al.* paper is important in trying to address whether sheep can transfer learning to different views of the same individual human, since this would imply individual rather than picture-based discrimination. However, it is not the first study to demonstrate this general ability in sheep. It was first reported that sheep could successful transfer learning of sheep face discriminations from frontal to profile views without additional learning in 2001 [10] and subsequently another group were also able to demonstrate successful transfer of learning between views of the faces of conspecifics both at different ages and different orientations (frontal turned to profile) [11]. In terms of human faces, a previous study also showed that sheep were able to recognize a specific familiar stockman from their face picture and exhibited an emotional response (vocalization) to it even after not having seen this individual for over a year [8]. Of note too is that in initial behavioural experiments in sheep where no training was given, animals demonstrated the ability to discriminate between pairs of faces with differential attraction, which also suggests that they treated the pictures as representative of real individuals [12]. Indeed, mere exposure to a picture of a sheep face, but not that of a goat, could reduce behavioural, endocrine and neural effects of isolation stress [13]. There is even some neural evidence to support the ability of sheep to form mental images of faces [10].

The issue raised about relative speeds of learning face associations and accuracies is relevant but not that surprising, given that humans are extremely quick and accurate at learning many different types of discriminations, whereas other species, including monkeys, may often require huge numbers of trials to learn the same thing. Of note, however, is that sheep learn to discriminate faster between sheep faces (either familiar or unfamiliar) than between simple geometric symbols and are also faster at learning to discriminate between the faces of familiar as opposed to unfamiliar sheep [14]. Even the most highly trained sheep using operant choice strategies (panel pressing) generally take at least 20–40 trials to achieve a greater than 80% accuracy criterion for discriminating between new pairs of faces [8,15], and humans are clearly able to achieve the same accuracy in fewer trials. This in itself, however, does not necessarily imply that humans have a qualitatively different expertise in learning faces, it could simply just be a more developed and efficient system for recognizing faces.

In summary, there is strong evidence that sheep do possess an expert system for recognizing the faces of individual sheep, and to some extent humans, and that this reflects recognition of the individual as opposed simply to a complex visual image. There is little doubt that humans are better than sheep in learning to recognize human faces, although our expertise in recognizing sheep faces is clearly more closely rivalled by sheep. From an evolutionary standpoint, it seems unlikely that sheep and human face-recognition systems in the temporal lobe differ markedly from a qualitative point of view, although clearly the human system is more developed and sophisticated.

Data accessibility. This article has no additional data.

Competing interests. I have no competing interests.

Funding. I received no funding for this study.

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
