## [Reviewer comments · Royal Society Open Science]

Review History

RSOS-182187.R0 (Original submission)

Review form: Reviewer 1

Is the manuscript scientifically sound in its present form?

Yes

Are the interpretations and conclusions justified by the results?

Yes

Is the language acceptable?

Yes

Is it clear how to access all supporting data?

Not Applicable

Do you have any ethical concerns with this paper?

No

Have you any concerns about statistical analyses in this paper?

No

Recommendation?

Accept as is

Comments to the Author(s)

Not Applicable

Review form: Reviewer 2 (Sebastian McBride)**Is the manuscript scientifically sound in its present form?**

Yes

Are the interpretations and conclusions justified by the results?

Yes

Is the language acceptable?

Yes

Is it clear how to access all supporting data?

Yes

Do you have any ethical concerns with this paper?

No

Have you any concerns about statistical analyses in this paper?

No

Recommendation?

Accept as is

Comments to the Author(s)

Dialogue, debate and critical discussion are fundamental to the scientific process. Professor Kendrick carried out the original and more expansive research on facial recognition in sheep and thus is very well placed to make critical comment on the work of Knoll et al. 2017 and also the response by Towler et al. His comments both contextualise the recent research by Knoll et al and also critically evaluates the validity of the Towler et al. response in terms of the prior research that has been carried out in this area. This greatly adds to the discussion identifying our current state of knowledge on the cognitive attributes of a non-human species such as the sheep.

Decision letter (RSOS-182187.R0)

07-May-2019

Dear Dr Kendrick:

It is a pleasure to accept your manuscript entitled "Response to "Sheep recognize familiar and unfamiliar human faces from two-dimensional images"" in its current form for publication in

Royal Society Open Science. The comments of the reviewer(s) who reviewed your manuscript are included at the foot of this letter.

on behalf of Dr Rosalind Arden (Associate Editor) and Dr Kevin Padian (Subject Editor).

Associate Editor Dr Rosalind Arden Comments to Author:

Associate Editor: 1

Comments to the Author:

Both reviewers concur that this ms should be accepted in its present form. One reviewer comments that this piece provides useful context for thinking about the topic. Thank you for the contribution.

Reviewer(s)' Comments to Author:

Reviewer: 1

Comments to the Author(s)

Not Applicable

Reviewer: 2

Comments to the Author(s)

Dialogue, debate and critical discussion are fundamental to the scientific process. Professor Kendrick carried out the original and more expansive research on facial recognition in sheep and thus is very well placed to make critical comment on the work of Knoll et al. 2017 and also the response by Towler et al. His comments both contextualise the recent research by Knoll et al and also critically evaluates the validity of the Towler et al. response in terms of the prior research that has been carried out in this area. This greatly adds to the discussion identifying our current state of knowledge on the cognitive attributes of a non-human species such as the sheep.
